# Reverse Sneezing in Dogs: Observational Study in 30 Cases

**DOI:** 10.3390/vetsci9120665

**Published:** 2022-11-29

**Authors:** Jesús Talavera, Patricia Sebastián, Giorgia Santarelli, Ignacio Barrales, María Josefa Fernández del Palacio

**Affiliations:** 1Division of Veterinary Cardiology-Pulmonology, Veterinary Teaching Hospital, University of Murcia, 30100 Murcia, Spain; 2Department of Veterinary Medicine and Surgery, Faculty of Veterinary Medicine, University of Murcia, Campus Espinardo, 30100 Murcia, Spain

**Keywords:** upper airway, canine, respiratory disease, nasopharynx, aspiration reflex

## Abstract

**Simple Summary:**

Reverse sneezing (RS) is a reflex triggered by nasopharyngeal irritation, which manifests as a paroxysm of loud inspiratory noise accompanied by a labored respiratory effort. It constitutes a common cause of consultation, as pet owners often worry and find these episodes to be stressful situations. However, no studies about the epidemiology, natural history, underlying disorders, or clinical response to treatment have been reported. The availability of specific data about all the mentioned aspects could contribute to a better definition of the importance of this clinical sign, in order to define the most appropriate diagnostic protocol, as well as the prognosis and the expected evolution. Signalment, clinical features, final diagnosis, and evolution were retrospectively studied in a cohort of 30 dogs with RS. Small and toy breeds may be predisposed to RS, but no predilection for sex, neuter status, or age exists. Many dogs continue to present RS despite treatment. Although some dogs present infrequent episodes of RS, being otherwise normal, RS should be considered a marker of potential irritation of the nasopharyngeal mucosa and should be sufficiently investigated.

**Abstract:**

Reverse sneezing (RS) is a frequent reason for veterinary consultation, but there is scarce clinical information. The aim of this study was to describe clinical characteristics in a cohort of 30 dogs with RS. Signalment, clinical features, results of diagnostic tests, final diagnosis, and evolution were retrospectively evaluated. Sex and neuter status were equally distributed into diagnosis categories. A significantly higher representation of toys (<5 kg, 50%) and small-sized dogs (5–15 kg, 27%), in comparison to medium (15–30 kg, 17%) and large-sized dogs (>30 kg, 7%), was found. RS was the main owner concern in many of the cases (67%). Many cases presented chronic RS (60%, > 3 months), with more than one episode a week (60%). Most cases had an additional clinical respiratory sign (63%) and an unremarkable physical examination (63%). Inflammatory airway disorders were present in 57% of the cases, followed by anatomical–functional disorders (27%), and nasal/nasopharyngeal foreign bodies (10%). Two dogs (7%) remained as open diagnoses. Episodes of RS were persistent despite the treatment in 61% of the dogs with follow-up. Although some dogs manifest infrequent episodes of RS, being otherwise normal, RS should be considered a marker of potential irritation of the nasopharyngeal mucosa and should always be sufficiently investigated.

## 1. Introduction

Reverse sneezing (RS) is defined as a mechanosensitive aspiration reflex triggered by nasopharyngeal irritation, which manifests as a paroxysm of loud inspiratory noise accompanied by a labored respiratory effort [1,2,3]. This paroxysm usually lasts a few seconds, and affected animals tend to adopt an orthopneic position, consisting of neck extension and elbow abduction. It is considered as a benign event that does not disturb respiratory function or pose a risk to life [1,2,3]. However, it constitutes a common cause of consultation, as pet owners often worry and find these episodes to be very stressful or potentially risky situations. Dogs are much more commonly affected than cats, although available pathophysiological studies have mainly been conducted in the latter species [4,5].

As an innate reflex, RS may appear spontaneously with a variable frequency in healthy animals, being considered idiopathic, nonprogressive, and more prevalent in small dogs [6]. If the frequency or severity increases, it may be indicative of a nasopharyngeal disease. Reported causes include excitement, foreign bodies, masses, nasal mites (i.e., *Pneumonyssus caninum*), viral infections, and epiglottic entrapment of the soft palate [1,2,3,7,8]. In such cases, the diagnostic protocol often includes imaging tests, such as radiography, endoscopy, fluoroscopy, and computed tomography.

Despite RS being a relatively frequent complaint in canine medicine, scarce and dispersed clinical information is available. To the authors’ knowledge, no studies about the epidemiology, natural history, prevalence of the underlying pathology, and clinical response to treatment have been reported. Brief information on RS can be found in specific chapters of veterinary medicine manuals [1,2,3,6,8] or as a secondary result in clinical studies on upper respiratory tract diseases [9,10,11,12,13].

The availability of specific data about all the mentioned aspects could contribute to a better comprehension of the importance of this clinical sign. In addition, it would be helpful to define the most appropriate diagnostic protocol, as well as the prognosis and the expected evolution.

The purpose of this retrospective study was to describe the signalment, clinical presentation, severity of RS, concurrent medical disorders, imaging findings, and evolution/response to treatments of a population of client-owned dogs with RS.

## 2. Materials and Methods

### 2.1. Case Selection

Medical records and complementary exam findings of dogs presenting with RS between February 2006 and July 2020 were reviewed retrospectively. The initial search was launched from the database of the hospital using the confirmed presence of RS episodes as the inclusion criteria, established in one of the following ways: by direct observation when the animal had an RS episode during the consultation, by visualization of a video recording of the event (provided by the owner), or by recognition of the clinical sign in example videos provided by the clinicians (Appendix A). All the cases with confirmed RS were initially included, and it was recorded whether the RS was the chief complaint for the consultation, or whether it was part of a broader list of clinical signs identified during the anamnesis. Later, only cases in which the clinical investigation had included head and neck radiographs, airway endoscopic evaluation, cytological or histopathological studies, and/or bacteriological culture results (if indicated) were definitively included.

### 2.2. Review of Medical Records

Medical records were reviewed, and cases were definitively included if the medical information was complete enough to retrieve the following data that were recorded: age, body weight, breed, sex, neuter status, clinical signs, frequency and duration of RS, pertinent physical examination findings, results of diagnostic tests, final clinical diagnosis, treatment, and evolution (when available).

The frequency of RS episodes was classified according to the following scale: very frequent (≥1 episode per day), frequent (≥1 episode per week but <1 episode per day), and infrequent (<1 episode per week). The episodes were also classified based on their duration as follows: acute (≤15 days), recent (15 days to 3 months), short-term (3–6 months), and long-term (≥6 months).

Endoscopic evaluation routinely included a pharyngo-laryngoscopy, posterior rhinoscopy, and tracheobronchoscopy, performed using a flexible videoendoscope (Fujinon EB-410S, Onys, Fujifilm, Tokyo, Japan) under anesthesia. When nasal involvement was suspected, anterior rhinoscopy was also performed using a rigid endoscope with a 2.7 mm diameter (Karl Storz SL 30° BA 64029BA; Karl Storz Endoscopy, Tuttlingen, Germany). Specimens for cytological or histopathological studies and/or bacteriological culture were taken from pharyngeal areas either under direct visualization (for example, the oropharynx), via rhinoscopy (direct and/or posterior), or by bronchoalveolar lavage according to the individual case. Cytology specimens were obtained using a brush or through fine-needle aspiration.

The reports of the radiological studies carried out in the nasal cavities, pharyngeal and laryngeal area, neck, and thorax were also included.

The final clinical diagnosis for each case was established by combining the history, physical examination findings, and results of complementary exams. The cases were grouped as follows: “foreign body”, “airway inflammatory disorder” (i.e., nasopharyngitis, rhinitis, laryngitis), and “anatomical-functional disorder”, which included diseases that directly affected the nasopharyngeal area (i.e., pharyngeal stenosis/collapse), but also diseases affecting the upper airways such as a collapsed cervical trachea. An “open diagnosis” was given to those cases in which diagnostic investigations did not identify a specific disorder.

The treatments implemented in each case were also extracted from the medical records and were later grouped for the study into corticosteroids, corticosteroids plus antibiotics, and others. Follow-up data were obtained from medical records or through owner interviews and included information regarding the resolution of the RS episodes. Cases were then classified according to whether the follow-up data were obtained less than or more than 6 months after the presentation. Where appropriate, the persistence of RS events and their frequency, using the scale described above, were recorded.

### 2.3. Statistical Analysis

Age and body weight are presented as the mean ± standard deviation. Baseline descriptive statistics were calculated and are reported as percentages for categorical data.

## 3. Results

### 3.1. Population

The initial search in the database provided 114 cases that were assessed for eligibility. Twenty-nine of them were excluded because either the presence of RS could not be confirmed by any of the methods described above (nineteen cases) or the information in the medical record was not complete enough (ten cases). Later, an additional 55 dogs had to be excluded because they had not undergone an endoscopic examination of the airway as part of the clinical investigation of RS.

Thus, a total of 30 dogs met all the inclusion criteria. The mean age was 5.1 ± 3.6 years (5 months to 12 years). A total of 53% (16/30) were females, and of these, 38% (6/16) were sexually intact; a total of 47% (14/30) were males, and of these, 57% (8/14) were sexually intact. The mean body weight was 9.9 ± 10.1 kg (1.1 to 36 kg). Sex and neuter status were equally distributed into diagnosis subgroups. However, the distribution of the population by body size was asymmetric, having a higher proportion of toys (<5 kg, 50%, 15/30) and small-sized dogs (5–15 kg, 26.7%, 8/30), in comparison to medium (15–30 kg, 16.7%, 5/30) and large-sized dogs (>30 kg, 6.7%, 2/30) (Table 1).

The study population comprised dogs of different breeds, including six crossbreed dogs (20%), five Yorkshire Terriers (17%), four Chihuahuas (13%), three Maltese dogs (10%), two Shih Tzus (7%), and one each of Poodle, West Highland White Terrier, Catalonian Shepherd, Golden Retriever, Labrador Retriever, Fox Terrier, German Shepherd, Spanish Hound, Pitbull, and Siberian Husky.

### 3.2. Clinical Presentation

RS was the main owner concern in the majority of the cases (66.7%, 20/30). Most of the cases had, in addition to RS, some other clinical respiratory sign (63.3%, 19/30), including: coughing (10/19, 52.6%), exercise intolerance/dyspnea (8/19, 42.1%), sneezing (6/19, 31.6%), nasal discharge (7/16, 36.8%), and respiratory noises (4/16, 25%). Digestive signs were present in 6/30 cases (20%), including gagging (2/6, 33.3%), vomiting (2/6, 33.3%), dysphagia (1/6, 16.7%), and hypersalivation (1/6, 16.7%). In 10 dogs, RS was the only clinical sign (10/30, 33.3%).

The global distribution of cases based on the time evolution of RS was asymmetric (Table 2), with a higher percentage presenting chronic symptoms (more than 6 months, 12/30, 40%); in 20% (6/30) of the cases, RS was present between 3 and 6 months before the presentation; in 23.3% (7/30) of the cases, RS was present between 15 and 90 days before the presentation, and only one dog (1/30, 3.3%) had an acute presentation. Information about the time evolution was not available for the other four cases (4/30, 13.3%). The distribution of cases based on the time evolution of RS by the different diagnosis subgroups is shown in Table 2.

Regarding the frequency of RS episodes (Table 2), most of the dogs had a high frequency of RS episodes. Therefore, 8/30 (26.7%) suffered RS episodes very frequently, 10/30 (33.3%) frequently, and 5/30 (16.7%) infrequently. This information was not available for the other seven cases (7/30, 13.3%).

The physical examination was unremarkable in the majority of cases (19/30, 63.3%). Four of thirty cases (13.3%) presented abnormalities considered unrelated to the RS origin. In the other seven dogs (23.3%), the physical examination findings suggested an upper respiratory disease, including increased reactivity to pharyngolaryngeal palpation (5/7, 71.4%), nasal discharge (3/7, 42.3%), and nasal flow reduction (2/7, 28.6%).

### 3.3. Diagnostic Investigations

In most of the cases (26/30, 86.7%), neck/cervical radiographs were normal (18/30, 60%) or showed abnormalities considered unrelated to the RS origin (8/30, 26.7%). Only in four cases did the neck radiographs show a reduction in the cervical tracheal diameter compatible with the cervical tracheal collapse that was considered probably related to the RS episodes when no other abnormalities were identified in the ancillary tests (Figure 1). Thoracic radiographs were normal (21/30, 70%) or showed a diffuse bronchointerstitial pulmonary pattern (9/30, 30%). In one dog, a gastric foreign body was found following the radiographic study that was related to gastroesophageal reflux and secondary pharyngitis inducing RS.

General blood test findings (hematology and biochemistry) were unremarkable in most of the cases (23/30, 76.7%) or revealed unrelated findings (7/30, 23.3%). Mild changes in the total and/or differential leukocyte count were the most common abnormalities.

The most common endoscopic finding was the presence of pharyngolaryngeal (including nasopharyngeal) congestion and edema (17/30 cases, 56.7%), followed by the presence of inflammatory follicles in the nasopharynx (9/30, 30%, Figure 2). Other findings included: hyperdynamic/spasmodic pharyngeal muscles (6/30, 20%), cervical tracheal collapse (5/30, 16.7%), elongated/flaccid soft palate (3/30, 10%), bilateral arytenoid stillness (3/30, 10%), tracheobronchial hyperemia and mucus accumulation (2/30, 6.7%), choanal (2/30, 6.7%, Figure 3) or nasal (1/30, 3.3%) foreign bodies, and choanal atresia (1/30, 3.3%). Endoscopy was considered normal in two dogs with RS (6.7%).

Brush cytology samples were obtained from the pharynx/nasopharynx in 15 dogs (50%), being classified as inflammatory lymphoplasmacytic infiltrate. Bronchoalveolar lavage fluid was obtained in four cases (13.3%), with two of them being classified as chronic lymphocytic, and the other two as eosinophilic inflammation. The bacterial culture of the bronchoalveolar lavage fluid of these dogs was negative in three and positive in one (*E. coli*). Biopsy samples of nasal (6/30, 20%) or pharyngeal (2/30, 6.7%) mucosa were compatible with mixed inflammatory infiltrates.

### 3.4. Final Diagnosis

Among all the 30 cases, most of them were diagnosed as suffering from an airway inflammatory disorder (17/30, 56.7%), mainly nasal/pharyngeal/nasopharyngeal inflammation (15/30, 50%), followed by eosinophilic bronchitis (2/30, 6.7%). Upper airway inflammation could be related to an upper digestive disorder in three dogs that had digestive symptoms (vomiting in two dogs, with one of them having a gastric foreign body, and regurgitation in one dog).

In eight cases (8/30, 26.7%), the presence of an anatomical–functional disorder was constated, including an elongated soft palate (3/30, 10%), cervical tracheal collapse (2/30, 6.7%), pharyngeal hypersensibility and collapse (2/30, 6.7%), and choanal atresia (1/30, 3.3%). In three dogs with RS (10%), the final diagnosis was the presence of choanal (2/30, 6.7%) or nasal (1/30, 3.3%) foreign bodies. A specific diagnosis could not be determined despite performing ancillary tests on two dogs, being classified as open diagnoses.

### 3.5. Treatment and Follow-Up

Oral and inhaled corticosteroids were the main treatment protocols used (23/30, 76.7%), either as the only treatment (11/30, 36.7%) or associated with antibiotics (12/30, 40%). The different treatments and the persistence of the RS episodes after treatment are detailed in Table 3.

Information about the follow-up of cases after diagnosis was available in 23/30 cases (76.7%), 16 of them through review of the medical record (16/23, 69.6%) and 7 of them through owner interviews (7/23, 30.4%). In 4/23 cases (17.4%), the follow-up data were obtained less than 6 months after the presentation, while in 19/23 (82.6%), they were obtained 6 months after the presentation.

The prescribed treatment resulted in the resolution of the RS episodes in nine cases with follow-up (9/23, 39.1%). Two of them underwent extraction of a foreign body (2/23, 8.7%), and the rest received corticosteroids alone (4/23, 17.4%) or with antibiotics (3/23, 13%).

Episodes of RS were persistent despite the treatment in 14/23 dogs with follow-up and even continued to be very frequent (5/23, 21.8%) or frequent (5/23, 21.8%), for diagnoses of both an airway inflammatory disorder (8/23, 34.8%) or an anatomical–functional disorder (5/23, 21.7%).

Between the two dogs with open diagnoses, one of them did not receive treatment and was lost to follow-up; the second one received oral and inhaled corticosteroids, but such treatment did not modify his RS frequency (more than one weekly episode) in the following year of follow-up.

## 4. Discussion

To the best of the authors’ knowledge, this is the first clinical study that includes a cohort of canine patients with RS. This cohort of dogs included various degrees of severity and persistence of this clinical sign in the presence of several respiratory disorders, whose causal relationship with RS is not always easy to establish (i.e., cervical tracheal collapse). It is worth mentioning that about 40% of the dogs presented a long-term history of RS as an isolated frequent clinical sign. Reverse sneezing is considered a respiratory reflex that has been described both in humans and in other mammalian species, including dogs and cats [14]. It was first reported in dogs by Tomori et al. in 1977 [15], but physiological studies have been carried out mainly in cats [4,5,14]. Experimentally, it has been shown that stimulating the nasopharynx mechanically and electrically induces a primary reflex called the gag-like aspiration reflex or sniff-like aspiration reflex (AspR), which is characterized by a short spasmodic inspiration not followed by active expiration [4,5,14]. The AspR is a rapid and strong spasmodic inspiratory effort accompanied by upper airway dilation, caused by stimulation of the phrenic and superior laryngeal nerves, and bulbar inspiratory neurons [16].

Veterinary medicine textbooks have traditionally described RS as a spontaneous manifestation of this AspR reproduced in experimental studies [1,17,18]. However, there are no studies in dogs with spontaneous RS that have clearly established whether RS and AspR are exactly the same. It is surprising that most studies have been conducted in cats, a species in which RS is not usually described as a spontaneous clinical sign. The AspR described in experimental studies is a primary reflex present even after medullary transection, and its main attributed function is self-resuscitation through the generation of agonic breaths [14]. Video recording studies of the glottis with parallel measures of pleural pressure (Ppl) in cats showed that the AspR is characterized by maximal contraction of the inspiratory muscles with simultaneous inhibition of expiratory muscles and rapid glottal opening. These activities tend to supply oxygen to the lungs and enhance the venous return to the heart. They support the perfusion of the myocardium and brain, preventing an imminent loss of consciousness and promoting auto-resuscitation [14,16]. Therefore, the AspR tries to generate the maximum negative Ppl, avoiding the collapse of the upper airway as much as possible. This does not seem to be compatible with the typical loud and noisy manifestations of spontaneous canine RS, during which loudness paroxysms necessarily imply dynamic stenosis of the pharynx/nasopharynx by spasmodic contraction–relaxation of the subsidiary muscles. All but two dogs (open diagnoses) in this study showed some alteration in the diagnostic examination that could have a causal relationship with RS. Thus, although RS may be, in origin, related to the AspR reproduced in experimental studies, it should be considered a defensive reflex secondary to irritation of the nasopharyngeal mucosa and not a part of the respiratory physiology in dogs with infrequent signs. Future studies using protocolized video recordings from orthogonal perspectives could help to characterize spontaneous canine RS versus the AspR described in experimental studies.

This study identified a group of dogs for whom RS was the main presenting clinical sign accompanying a variety of respiratory disorders. No significant predilection for sex (spayed or not) or age was found. However, as in previous studies evaluating pharyngeal disorders [13,19], many dogs were of a small size (under 15 kg). This could be due to the inherent peculiarities in the clinical population that presents to a referral hospital, as in this study. Another explanation for the higher proportion of small breeds is that, in the geographical area where this study was performed, these types of dogs tend to live inside the house and, in general, have more interaction with the owners, which makes it more likely for them to identify RS episodes. In contrast, medium and large breed dogs tend to live outside, so intermittent/infrequent signs might be unnoticed. Conducting studies to investigate the prevalence of RS in the general canine population could help to establish whether physiological RS is truly more prevalent in small breeds. In addition, the fact that small breed dogs have a relatively smaller pharyngolaryngeal region could also play a role in the higher prevalence of RS episodes in those breeds. In fact, small breeds are also more prevalent in studies of pharyngeal [13,19] and laryngeal disorders [20].

To have consistent clinical data, this study included only dogs who had undergone a complete clinical evaluation oriented to investigate the RS origin which included chest and neck radiographs and endoscopic evaluation of the airways. This determined the exclusion of 55 dogs suffering from confirmed RS that had been attended to in consultation. These data are not included in the study, but it is worth mentioning that the more frequent profile for these dogs was small size or toy, without other clinical signs, and with a long-term history of RS (>6 months), a low frequency of episodes (less than 1/6 months), and a normal physical examination (or unrelated findings). These dogs were typically not put under treatment. Obviously, this defines a dog for which the RS was interpreted as probably irrelevant by the clinician; therefore, diagnostic testing was not considered mandatory or urgent. It is thought that this reflex spasm of the nasopharyngeal and soft palate musculature aids in the transportation of mucus from the caudal nasal passages and nasopharynx towards the oropharynx, where it can be swallowed and removed from the airways [17,18]. This function has also been suggested in experimental studies on the AspR [14], and to the authors, it seems much more in line with the clinical context in which RS episodes are identified in dogs. Its presence in dogs with nasopharyngeal disorders is logical considering the anatomic location of the problem. In the case of animals with a low frequency of RS episodes, an anatomical predisposition (especially in small breeds) and/or a deficit of focal mucociliary function could be indicated. The possibility of performing dynamic contrast studies through fluoroscopy during an episode of spontaneous RS would help to clarify this hypothesis. Unfortunately, this type of study is difficult to perform since a protocol for the non-invasive stimulation of RS is not available.

Reverse sneezing is considered as a specific sign of nasopharyngeal, caudal nasal, or sinus disease [1,2,3,7,8], but there are no specific data on the relative prevalence of the different causes. In the present study, airway inflammatory disorders were the most frequently diagnosed (57%), followed by anatomical–functional disorders (27%), while the presence of foreign bodies was infrequent (10%). Upper airway inflammation could be related to an upper digestive disorder in three dogs that had digestive symptoms (vomiting in two dogs, with one of them having a gastric foreign body, and regurgitation in one dog). Therefore, gastrointestinal disorders should be considered when concurrent vomiting, gagging, and dysphagia are present. The cervical tracheal collapse was identified in 2/30 dogs, a finding that has not been previously reported. The dynamic collapse of the airway perpetuates additional inflammation, tracheal edema, alterations or failure of the mucociliary apparatus, increased mucus secretion, and mucus trapping within the airways [21]. A hypothesis would be that this inflammatory context in combination with the increased respiratory secretions may predispose these dogs to RS. However, it must be considered that the association between the presence of RS and the diagnosis of tracheal collapse could also be incidental rather than causal.

The presence of masses is listed among the possible causes of RS [1,2,3,7,8]. However, RS does not usually appear in the list of clinical signs caused by nasal tumors [22,23]. Nasopharyngeal masses can be expected to induce RS in affected patients since they can constitute a source of nasopharyngeal irritation. In this study, nasal, nasopharyngeal, or sinus masses were not identified in dogs with RS.

Nasal mite infestations (*Pneumonyssoides caninum*) have also been related to RS, and these parasites are widely distributed, having been reported in Northern Europe, the Middle East, and North America [1,2,3,7,8]. However, nasal mite infection was not identified in any dog in the present study. Certainly, to the best of the authors’ knowledge, the presence of this parasite has not been communicated in the geographic area where the study was performed. However, even in areas with a high prevalence of nasal mites, a low morbidity rate has been described [24]. This may suggest that the mite resides in many dogs and is not associated with either pathological or clinical changes. Thus, this infection could be underdiagnosed.

The major limitations of this study are associated with its retrospective nature. Although well-completed medical records were selected and those with little or confusing information were discarded, a certain degree of subjectivity in the collection of data and their verification in the medical records is inevitable. Additionally, information on each variable was not available in 100% of the cases. Despite these limitations, the data provided constitute a “still photo” of the presentation of this clinical sign in a clinical cohort of canine patients that can constitute objective knowledge, which could serve as a guide and support for clinical decision-making in dogs that suffer from RS. Ideally, further prospective studies in a larger population could help to identify differences that may have gone unnoticed in this study.

## 5. Conclusions

Small and toy breeds may be predisposed to present RS, but no predilection for sex, neuter status, or age exists. Airway disorders (inflammatory and anatomical–functional) are frequently present. Many dogs continue to suffer from RS despite treatment. Although some dogs present infrequent episodes of RS, being otherwise normal, RS should be considered a marker of potential irritation of the nasopharyngeal mucosa and should always be sufficiently investigated.

## Figures and Tables

**Figure 1 vetsci-09-00665-f001:**
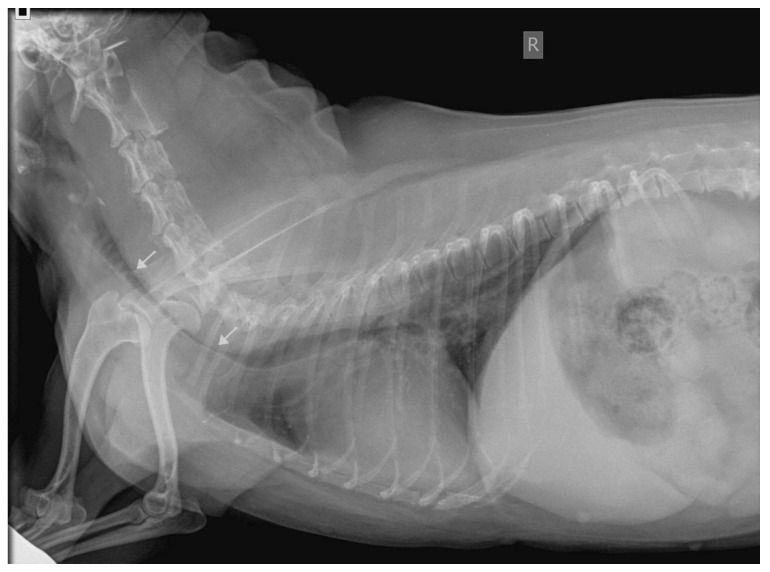
Laterolateral radiographic view of the neck and thorax of a dog with reverse sneezing showing a dorsoventral tracheal collapse that affects the cervical portion of the trachea up to the thoracic inlet (arrows).

**Figure 2 vetsci-09-00665-f002:**
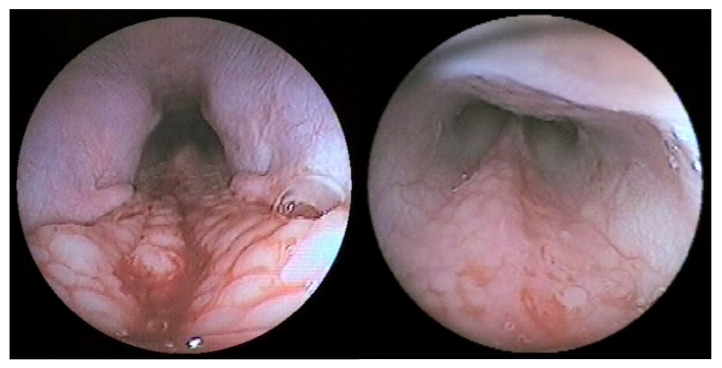
Endoscopic images of the nasopharynx of a dog with reverse sneezing showing extensive follicular inflammation, both in the caudodorsal portion of the nasopharynx (**left**) and extending rostrally to the end of the nasal septum (**right**).

**Figure 3 vetsci-09-00665-f003:**
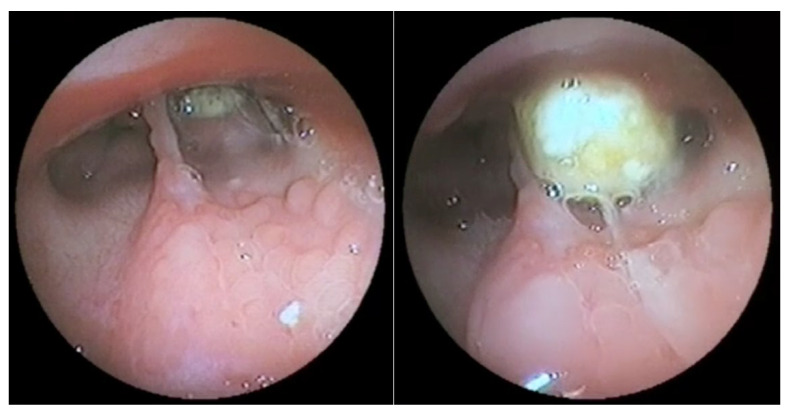
Endoscopic features of the nasopharynx of a dog with reverse sneezing showing the presence of a foreign body lodged in the right choana before (**left**) and after (**right**) its partial mobilization by flushing. Note the presence of follicular inflammation of the surrounding mucosa. In this case, the foreign body (dry chickpea) was removed by pushing it with the help of forceps introduced through the nostril, which resolved the reverse sneezing.

**Table 1 vetsci-09-00665-t001:** Signalment by the main diagnosis subgroups.

		All (N = 30)	Inflammatory Disorders (N = 17)	Anatomical–Functional Disorders (N = 8)	Foreign Bodies (N = 3)	Open Diagnosis (N = 2)
Age (months)		61.2 ± 42.6 (5–144)	65.2 ± 37.7 (18–132)	49.6 ± 48.5 (7–126)	84 ± 60 (24–144)	47.4 ± 33.5 (5–72)
Weight (kg)		9.9 ± 10.1 (1.1–36)	13.2 ± 11.5 (1.1–36)	4.2 ± 2.3 (2–9)	12.5 ± 10.2 (4.6–24)	1.8 ± 0.3 (1.6–2)
Sex	Male	46.7% (14/30)	58.8% (10/17)	37.5% (3/8)	33.3% (1/3)	-
	Female	53.3% (16/30)	41.2% (7/17)	62.5% (5/8)	66.7% (2/3)	100%
Size	Toy (<5 kg)	50% (15/30)	35.3% (6/17)	52,9% (9/17)	33.3% (1/3)	100%
	Small (5–15 kg)	26.7% (8/30)	29.4% (5/17)	47.1% (8/17)	33.3% (1/3)	-
	Medium (15–30 kg)	16.7% (5/30)	23.5% (4/17)	-	33.3% (1/3)	-
	Large (>30 kg)	6.7% (2/30)	11.8% (2/17)	-	-	-

**Table 2 vetsci-09-00665-t002:** A breakdown of reverse sneezing (RS) characteristics by the different diagnosis subgroups.

		All (N = 30)	Inflammatory Disorders (N = 17)	Anatomical–Functional Disorders (N = 8)	Foreign Bodies (N = 3)	Open Diagnosis (N = 2)
History of RS at presentation	≤15 days	3.3% (1/30)	5.9% (1/17)	-	-	-
	15–90 days	23.3% (7/30)	41.2% (7/17)	-	-	-
	3–6 months	20% (6/30)	5.9% (1/17)	25% (2/8)	66.7% (2/3)	50% (1/2)
	≥6 months	40% (12/30)	41.2% (7/17)	50% (4/8)	-	50% (1/2)
	Not available	13.3% (4/30)	5.9% (1/17)	25% (2/8)	33.35 (1/3)	-
RS frequency at presentation	≥1/day	26.7% (8/30)	35.3% (6/17)	12.5% (1/8)	50% (1/2)	-
	≥1/week–<1/day	33.3% (10/30)	35.3% (6/17)	25% (2/8)	50% (1/2)	50% (1/2)
	<1/week	16.7% (5/30)	17.6% (3/17)	25% (2/8)	-	-
	Not available	23.3% (7/30)	11.8% (2/17)	37.5% (3/8)	-	50% (1/2)
RS persistence at follow-up	≥1/day	16.7% (5/30)	23.5% (4/17)	12.5% (1/8)	-	-
	≥1/week–<1/day	13.3% (4/30)	5.9% (1/17)	25% (2/8)	-	50% (1/2)
	<1/week	13.3% (4/30)	1.2% (2/17)	25% (2/8)	-	-
	Resolution	30% (9/30)	35.3% (6/17)	-	100%	-
	Not available	26.7% (8/30)	23.6% (4/17)	37.5% (3/8)	-	50% (1/2)

**Table 3 vetsci-09-00665-t003:** Comparison by therapeutic subgroups of main diagnosis of reverse sneezing (RS) and persistence after treatment.

	Inflammatory Disorders (*N =* 17)	Anatomical–Functional Disorders (*N =* 8)	Foreign Bodies (*N =* 3)	Open Diagnosis (*N =* 2)	Persistence of RS (*N =* 14)	Disappearance of RS (*N =* 9)
No treatment	5.9% (1/17)	12.5% (1/8)	-	50% (1/2)	NA	NA
Glucocorticoids	23.6% (4/17)	75% (6/8)	-	50% (1/2)	35.7% (5/14)	44.4% (4/9)
Glucocorticoids + antibiotics	64.7% (11/17)	12.5% (1/8)	-	-	57.2% (8/14)	33.3% (3/9)
Others	5.9% (1/17)	5.9% (1/17)	100% (4/4)	-	7.1% (1/14)	22.2% (2/9)

NA, not available.

## Data Availability

Not applicable.

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
