# Peer review of "Reverse Sneezing in Dogs: Observational Study in 30 Cases"

_vetsci, 2022, doi:10.3390/vetsci9120665_

Round 1

Reviewer 1 Report

Reverse Sneezing in Dogs: Observational Study in 85 Cases

By Talavera et al.

This manuscript analyzes the occurrence of Reverse Sneezing (RS) in dogs by doing a retrospective study of the records from the Veterinary Teaching Hospital to check in which kind of patients it occurs (according to their weight/size, age, sex and neuter status) and if it appears associated with other pathologies. As in most cases RS is not a reflex that occurs very frequently in the consultation (since most patients have chronic symptoms, more than 6 months and they are mainly reported by their owners), this work is a first approach to its presentation, trying to analyze different aspects that concur in the analyzed patients.

The video from the supplementary material is very useful.

Major comments:

-Table 1: According to the text (L136-138), data labeled as spayed males and females (SM and SF), correspond to sexually intact males and females. In addition, in the “neuter status” section there are two categories: spayed and neutered, but no intact animals. Please, check.

-Table 3: According to this table, there is a mistake in L211-3, as it is said “Treatment was most often prescribed in dogs with persistent RS (16/35, 46%) than in those in which the RS disappeared (1/16, 6.3%) (P<0.05) (Table 3).”  Data 16/35 and 1/16 (columns E and F, respectively) correspond to the line of “No treatment”.

Minor comments:

L52: at the end of the sentence, the reference 6 should be as a superscript.

L54: species name should be in cursive.

L136 and L138: signs +/- or similar (to express mean +/- standard deviation) are missing.

L198: try to pass the heading 3.4 to the next page.

L208-9: remove capital letter of Group.

L220-3: remove capital letter of Group/s.

L228: Figure 1. Instead of “cranially” in the head, the correct term is “rostrally”.

L230-2: Figure 2. Maybe adding some arrows pointing to the tracheal collapse will be easier to localise it.

L235: Figure 3. The images are not labeled with A or B.

L279: Remove the capital letter from “Authors”.

L287: … in dogs by Tomori in 1977… As this article is authored by more than two authors, you should add “et al.”: … in dogs by Tomori et al. in 1977…

L425-6 (Reference #2) and L433-4 (Reference #6). Both references are referred to different chapters from the same book, although one is from the 8th ed. and another from the 7th ed. Would it be better to choose the most recent edition? As it is supposed that all the chapters are present and reviewed in the most modern edition.

Reviewer 2 Report

This original, retrospective study describes a relatively large series of canine patients with Reverse Sneezing (RS).

While this is an interesting attempt to describe epidemiology and underlying pathology several major and minor concerns must be addressed before publications.

My main complains are about the retrospective nature of this study; the main inclusion criteria is confirmed RS while endoscopy was performed in only 35% of cases, radiographs in 55% of cases and blood tests in half of cases. This makes the cohort of dogs highly nonuniform and this inevitably implies severe bias affecting both discussion and conclusions. To get better and sound results, inclusion criteria should include, in all cases, breed-age-sex-weight, head and neck radiographs, anterior rhinoscopy and flexible endoscope nasopharingoscopy, biopsy if indicated, concurrent respiratory condiotions. Bloodwork non necessary. The lack of these information lead to an unacceptable high number (49%) of "open diagnosis", that is an elegant way to indicate a "missed diagnosis".

Minor complains:

Line 157: how tracheobronchitis can be considered an airway inflammatory disorder causing RS?

Line 158: how tracheal collapse can be considered an airway condition causing RS? although you try to give an explanation to this, tracheal collapse and RS are concomitant events in toy breeds and small breed dogs;

Line 177: the foreign bodies are nasal or nasopharyngeal? Sneezing and nasal discharge are symptoms of nasal foreign bodies;

Line 197: how gingivostomatitis can be considered a RS-related finding?

Line 197: how a such non specific finding such as regional lymphatic (by the way, which ones?) nodules should be considered a RS-related finding?

Line 285: RS should be considered a defensive reflex secondary to irritation of nasopharyngeal mucosa and not a part of respiratory physiology in dogs with infrequent sign; this could lead to the wrong belief that occasional RS should not be sufficiently investigated;

Line 314: I've never heard nor never known about the presence of loud sounds during the expiratory phase of RS; could you please better specify where this have been described in Venker-van Haagen book?

Line 360: what do you mean with "parapharyngeal inflammation"?

Tables 1 and 2 are, in my opinion, confusing and doesn't help the reader in a better interpretation of results;

This just to highlight my main complains

Round 2

Reviewer 2 Report

Dear Authors,

I strongly appreciate you point by point answer but I need the text of the revised version of the paper with all the changes included; it's almost impossible and extremely time consuming to try to understand your changes in the version I received.

Author Response

Dear reviewer,
I apologize for not having attached the file with the changes included. Considering the major changes introduced, it seemed to us that the evaluation would be clearer without them. I attach here the requested version.
Best regards.

Round 3

Reviewer 2 Report

I really appreciate the changes you did in your paper; it seems to me much more readable and of higher quality if compared with the first writing. 

Before final acceptance some more minor corrections are neeeded:

Line 10: please, change (Germany) with (The Nederlands);

Line 172: please, specify identification codes for both the endoscopes you listed;

Line 213: Maltese Dogs;

Line 215: German Shepherd Dog;

Line 419: please, delete "(less than 15 days of duration)" as this is a redundant information; you already explained it in Material and method section;

Lines 423 - 425: please, delete redundant information (see comment at line 419)

Line 435: please, delete "consistent"

Line 437: please, delete "(see later)"

Line 446: "followed by"

Line 449: please, replace "laryngeal paresis" with "bilateral arytenoid stillness";

Line 820: "inflammation";

Line 839: please, replace "disappareance" with "resolution";

Line 840: please, replace "correspond to" with "followed the";

Line 939: please, replace "images" with "features";

Lines 1771 - 1773: the meaning of this sentence is unclear. Please, rewrite or delete; 

Line 1851: a nasopharingeal mass is almost always a cause of RS in dogs;

Lines 1915-1918: I suggest to delete these lines;

In addition, I suggest a thorough linguistic editing of this paper
